# Evaluating Successful Aging in Older People Who Participated in Computerized or Paper-and-Pencil Memory Training: The Memoria Mejor Program

**DOI:** 10.3390/ijerph16020191

**Published:** 2019-01-11

**Authors:** Carmen Requena, George W. Rebok

**Affiliations:** 1Faculty of Education, University of Leon, 24071 Leon, Spain; 2Department of Mental Health and Center on Aging and Health, Johns Hopkins University, Baltimore, MD 21218, USA; grebok1@jhu.edu

**Keywords:** older adults, computerized training, memory training program, EEG, successful aging, well-being

## Abstract

*Background.* The evaluation of successful aging includes objective criteria to measure cognitive function and psychological well-being and levels of functional capacity needed to perform daily activities related to the preservation of autonomy. In addition, the emergence of computerized cognitive training programs has allowed us to use a new class of tools to verify the theoretical postulates of neural plasticity in aging. *Objective.* The present study investigates subjective and objective criteria of successful aging in healthy older adults participating in a memory training program offered as two versions: computer and paper-and-pencil. *Method.* Fifty-four healthy older adult participants recruited for the study were organized into two training groups. Group 1 (G1) used the computer program and Group 2 (G2) used the paper-and-pencil program. *Results.* The analysis revealed no significant differences in psychological well-being between the two training groups. However, the groups did differ significantly in objective evaluations of successful aging, as measured by attention and everyday memory, and brain activity as measured by sLORETA, with G1 outperforming G2 on both measures. *Conclusion.* Computerized memory training programs show promise for restoring cognitive and cerebral functioning in older adults, and consequently, may be better suited to achieving the objective criteria of successful aging than paper-and-pencil memory training programs. However, this conclusion should be taken with caution since differences in age and educational level may have influenced the results.

## 1. Introduction

The evaluation of successful aging includes objective criteria to measure levels of functional capacity [1,2]. In addition, functional capacity must be sufficient to allow older people to have a social network and accomplish productive activities [3]. The social network covers socio-emotional needs (such as attachment and relationships with peers), and instrumental support in daily tasks [4]. The productive activities carried out by retired people are not evaluated in economic terms, but are looked at in terms of how these types of activities contribute to the maintenance of a healthy society and older people’s personal development [5,6]. On the other hand, a large group of researchers agree on the importance of including subjective parameters among the indicators of successful aging [7]. These indicators are more related to personal well-being, subjective assessment, or the meaning that the person themself attributes to well-being. Wellness is defined as the experience of happiness or satisfaction, challenges that require effort to overcome them, and the achievement of valuable goals [8]. However, the subjective criteria alone have been shown to be insufficient to preserve the quality of life because sometimes older people value unfavorable situations that move away from the objective standards of quality of life as “a good way of aging” [9].

The pursuit of fulfilling activities is one of the many concepts related to successful aging. Generally, the subjective keys to aging in a successful manner are related to the “personal evaluations” of the value or utility of the activities that older adults perform in their free time. In this sense, research shows that there are two indicators that are especially valued by older people. One refers to the social benefits that activities bring [10] and the other to the ability to freely choose an activity (self-determined motivation). Both subjective indicators correspond to the attainment of objective welfare standards. However, according to expert evaluation, other subjective motivations are based on fear of deterioration or on the experience of lack of meaning and purpose in daily activities as one ages [11]. Taking into account the objective criteria of successful aging proposed in the last 50 years, cognitive training programs have been developed that are expressly adapted to the normative assessments of quality of life established by literature. In particular, the aim of these programs is to preserve cognitive function and to maintain functional independence in healthy older adults until the end of their lives. Examples of the success of this type of program are the longitudinal programs ACTIVE (for Advanced Cognitive Training for Independent and Vital Elderly) [12] and Memoria Mejor [13]. These programs were developed in the late nineties with paper-and-pencil, and a decade later they were computerized, showing, among other advantages, the personalization of training [14]. It has been found that personalization determines individual adjustment to the difficulty of a task, which in turn increases the effectiveness of the training and improves its adherence [15,16]. On the other hand, computerized programs (compared to traditional methodologies) facilitate, increase, and preserve participants’ enthusiasm to continue in the program through innovative exercises [17,18], rewards associated with the accomplishment of the task, and progressively challenging activities [19]. In this case, these quantifiable subjective evaluations coincide with the objective criteria of successful aging.

In addition, the emergence of training programs for computers has allowed us to have a new class of tools to verify the theoretical postulates of neural plasticity, including plasticity in later life [20]. Therefore, the combination of neuroimaging techniques with computerized training programs has led to new theoretical approaches to cognitive and cerebral neuroplasticity in older people. For example, reverse hierarchy theory [21] bases the programs of cognitive stimulation in training focused on neural networks at the superior or cortical level. It is argued that superior regions of the brain are integrative and have the ability to amplify activation by selective disinhibition processes in cortical networks with which they communicate. Once training of the superior regions has succeeded in restoring the anterior/posterior and interhemispheric connection, not only is cerebral plasticity restored, but it is also possible to measure and quantify it. The literature has echoed not only these theoretical approaches about the increase in plasticity, but also programs that have been designed and developed to experimentally verify the measurable increase in plasticity, such as BrainHQ [22]. This program refutes the hypothesis that the destructive changes that are presaged in aging are inevitable. With the training of superior or cortical areas, deterioration can be delayed or reversed, and predetermined connections between the anterior and posterior brain can be reestablished [19].

In short, training programs in computer support designed to promote successful aging incorporate cognitive, emotional, and cerebral criteria [12,22]. Cognitive variables such as memory, attention, information processing, and reasoning are measured by standardized tests adapted to the age and educational level of the elderly [23]. Emotional variables are measured with scales of the quality of life that include emotional dimensions, such as self-acceptance and personal growth. Finally, brain variables are quantified by neuroimaging techniques that identify the restoration of “functional disconnection” caused by the weakening of feedback between cortical and subcortical circuits [20,24]. In addition, the recording of electrical activity during training allows the measuring of the evolution of neural plasticity in the trained areas. Although each of these variables has been described theoretically and empirically in the literature, there is no known study that has contrasted their evolution in an integrated way using a cognitive training program with computer and classic support (paper-and-pencil). The objective of this research is to evaluate subjective and objective criteria of successful aging in healthy older people participating in a training program offered as two versions: computer and paper-and-pencil. In particular, a psychological well-being scale will be used as a subjective criterion, and word memory and brain activity in electro-magnetic bands will be used as objective criteria.

## 2. Method

### 2.1. Paricipants

This is a retrospective study that evaluates older people who participated in the Memoria Mejor training program, a cognitive program using either paper-and-pencil or a computer. The training program, Memoria Mejor, was selected because it is available using both approaches. The sample originally consisted of 82 participants older than 65 years who chose the item “learn new skills and abilities” in the Cochran Baby Boomer Quiz [25] as their main reason for participating in the Memoria Mejor program. The study was carried out with those who participated in the program in the years 2015 to 2017 in the senior centers of Ponferrada (Spain).

Inclusion criteria: Participants in the Memoria Mejor training program who used both a paper-and-pencil and computerized version. The exclusion criteria were: a medical history of neurological or psychiatric disease, and MMSE <25/30 [26].

Of the original 82 people, 54 participated in the study; seven declined to participate, eight did not complete the evaluation, six could not be contacted, and the rest had a score lower than 25 on the MMSE. Participants recruited for the study were organized into two groups, according to the support with which they carried out the memory program. In particular, Group 1 (G1) carried out the computerized program and Group 2 (G2) completed the program using paper-and-pencil (see Table 1). This study was approved by the Ethics Committee of the University of León in 2017, and it was carried out following the Deontological Standards recognized by the Helsinki Declaration of 1975 (as revised in the 52nd Annual General Assembly in Edinburgh, Scotland, in October 2000), the standards of Good Clinical Practice and the Spanish Legal Code regulating clinical research involving human subjects (Royal Decree 223/2004 about regulation of clinical trials).

### 2.2. Process

Sample recruitment was carried out in four senior centers in Ponferrada, Spain. An informational talk was given about the objective of the study to older people who usually go to senior centers to do leisure activities. Older people interested in voluntarily participating in the study filled out a document with their name, surnames, age, and contact telephone number. Participants were scheduled for evaluations via telephone call.

There were two evaluation sessions. The first evaluation lasted approximately 18 min. The MMSE [26] and the Ryff Psychological Well-being Scale were administered [27]. The second evaluation was completed on another day in the morning and consisted of two assessments—an EEG record taken at baseline and the other while the participant performed a word memory task. For the EEG assessment, participants were asked to sit comfortably with their eyes closed and to try to be relaxed without tension in any part of their body. For the second assessment, participants were asked to try to remember as many words as possible, maintaining the same postural position while keeping their eyes open. The list of words was presented twice consecutively. After five minutes (the time it took to remove the EEG registration cap), participants were asked to say aloud the words they remembered.

#### 2.2.1. Memoria Mejor Training Program

The Memoria Mejor memory training program has both a paper-and-pencil and a computerized version [28,29]. The program is based on pastime exercises selected from journals and magazines by older people themselves. Exercises were designed to improve linguistic, numeric, spatial, and constructive skills. Pastimes are: (1) alphanumeric code; (2) extraction of words from other words; (3) word completion from missing vocals; (4) recognition of misplaced words; (5) alphabet soups; (6) peseta/euro conversion; (7) tangram (5 levels of difficulty); (8) dominos; (9) magic stair; (10) crosswords combined with labyrinths; (11) knight moves; (12) operations with addresses; (13) calculation of prices of fruit; (14) letter puzzles (one level of difficulty); and (15) colors and shape layout patterns. Pastimes are also ordered by levels of difficulty, with at least six individual exercises for each type of exercise.

The temporal distribution of training sessions in both groups was as follows: 32 sessions were held weekly in the senior centers for 75 min during the months of October to May during the years 2015–2017. The groups were organized into groups of between eight and 10 people. In total, 60% of the session time was set aside for modules and pastimes, 30% involved debate and discussion concerning the difficulty of the exercises and their daily life application, and 10% was dedicated to addressing questions raised by homework exercises which were repetitions of already trained abilities.

##### Therapeutic training

Eight psychologists were trained to administer the Memoria Mejor program by paper-and-pencil and on the computer according to the following parameters:

1. Basic Training.

This was done prior to the beginning of the research study and lasted 25 h. The Basic Training was delivered as an intensive course, both theoretical and practical, to address the following topics:

1.1. Memory: memory operations, phases and types of memory, memory disturbances, and strategies.

1.2. Memoria Mejor training program: session implementation and distribution of content per session and role-playing with voluntary older adults.

2. Monitored sessions

This was accomplished through continuous activities coordinated throughout the study involving regular contact of the research group with each Memoria Mejor trainer. This helped to improve and maintain the trainers’ skills, as well as to make the exercises by paper-and-pencil and on the computer more uniform.

#### 2.2.2. Measuring Instruments

The Ryff Psychological Well-being Scale [27] has been adapted to the Spanish population [30]. This multidimensional scale is an instrument that has 39 items in which participants respond between 1 (totally disagree) and 6 (totally agree). This rating scale meets subjective evaluation criteria of good psychological functioning on the basis of six dimensions or positive attributes of psychological well-being established by Ryff. The first dimension is self-acceptance or a positive attitude towards the self. This dimension refers to feeling good about yourself even when you are aware of your own limitations. Having positive attitudes towards oneself is the way to measure the positive psychological functioning of this dimension [31]. The second dimension refers to the ability to maintain positive relationships with other people. People need to maintain stable social relationships and have friends they can trust. A third dimension is the autonomy to maintain one’s own individuality in different social contexts. The fourth dimension refers to the domain of the environment. That is, the personal ability to choose or create favorable environments to satisfy one’s own desires and needs is another characteristic of positive functioning. A fifth dimension refers to the need to have purposes in life. One needs to define objectives that give meaning to life. Finally, the dimension of personal growth refers to the effort to develop one’s potential, to continue growing as a person, and to maximize one’s capabilities [32].

The Stroop Color task [33] is a test designed to measure attentional processes. The task, administered by paper modality, consists of six cards. The first two cards contain names of colors printed in an incongruent ink color (Incongruent condition). The third and fourth cards contain names of colors printed in a congruous ink color (Congruent condition).

Memory was evaluated through the standardized measure Rivermead Behavioral Memory-RBMT Test [34]. The RBMT is a battery designed to evaluate the participant’s memory while they do everyday tasks. There is evidence that favors the use of the RBMT in older adults and for the neuropsychological assessment of memory impairment [35]. The RBMT assesses different types of memory, such as associative memory, prospective memory, visual memory, verbal memory, topographic memory, control, and recognition strategies which produce a global score from 0 to 12 points.

#### 2.2.3. Word List from the Wechsler Memory Scale

The Word List of the Wechsler Memory Scale, third edition (WMS-III) [36], assesses the ability to learn a list of words by repetition. The Word List of the WMS-III consists of 12 words without semantic or phonetic relation that are read aloud to the participant. In our study, the list was read twice consecutively.

#### 2.2.4. EEG Recording and Analysis

The EEG was recorded with a 64-channel amplifier IC Neuronic S.L., City of Havana (Cuba) compose of: Control Unit (Model ND001M/Serial No.: 09AND001M-032) and amplifiers block (Model ND1400/Nº Serial: 09AND1400-003) and specific acquisition software EEG/Edition EEG N_E-SW-1.2/7.1.4.1. Reference electrodes were placed on the earlobes. In addition, electrooculography (EOG) was registered using three pairs of external electrodes in order to record the horizontal and vertical movement of the eyes. Electrode impedance was set for each participant before data collection, but always kept below 5 KΩ. The recording was carried out using an Special Order Cap (https://electro-cap.com/index.cfm/caps/#special) with Ag/AgCl electrodes, Neuronic S.L City of Havana (Cuba) which made it possible to analyze the active scalp areas of the subjects. ERP signals and stimulus markers were continuously recorded at a sampling frequency of 200 Hz during the presentation of the task. The signals were filtered using a band-pass finite impulse response filter with a Hamming window between 1 and 70 Hz. In addition, a 50 Hz notch filter was used in order to remove the power line artifact. Finally, an artifact rejection algorithm was applied to minimize oculographic and myographic artifacts [37]. In particular, components related to eye blinks, according to a visual inspection of the scalp maps and their temporal activations from independent component analysis (ICA), were discarded [38]. Thirty artifact-free segments were selected for each of the basal state conditions and for word memorization. The analysis of frequency bands: delta (0.5–4), theta (3.5–7.5 Hz), alpha (8–11.5 Hz), beta (12–19.5 Hz), was carried out with the Quantitative EEG Analysis (qEEG) software/No. N_I-SW-5/6.2.3.0, Neuronic S.L, City of Havana (Cuba) Neuronic S.L. Cuba). This program allows us to estimate the spectral activity in both topography and source localization.

#### 2.2.5. Source Localization

This technique has been widely used to study the neural correlates of cognition because it combines a high temporal resolution of the EEG technique with a reasonable spatial identification of the electrical signal of the cortical sources [39] (see http://www.uzh.ch/keyinst/NewLORETA/sLORETA/sLORETA.htm). The sLORETA software divides the brain into a total of 6239 cubic voxels with a resolution of 5 mm and estimates the density of the current source [40]. In the current investigation, the localization sources were estimated with an analysis of 64 electrodes located in the frontal, medial, temporal, and bilateral parietal regions. Participants were registered using the International System 10–20. The sources were calculated for every participant and each condition at the four bands of frequency with the Brain Cracker, City of Havana (Cuba), Neuronic S. L., which used low resolution electromagnetic tomography (Loreta implemented in sLoreta) [41]. The sLoreta source current density was calculated from scalp-recorded ERP using a realistic head model from the Montreal Neurological Institute (MNI) [40], in which the 3-D solution space was restricted to only the cortical gray matter [42]. The ERP voltage topographic maps were made by plotting color-coded isopotentials obtained by interpolating the voltage values between the scalp electrodes in specific latencies. Voxel wise non-parametrical statistics as implemented in sLORETA were used.

##### Statistical Analysis

Statistical treatment of the data was carried out using the IBM^®^ SPSS^®^ Statistics 17 (Chicago, IL, USA) The Kolmogorov Smirnov normality test was carried out with the Lilliefors correction for all the variables obtaining a value *p* > 0.05. The data were analyzed by means of a one-way analysis of variance (ANOVA) using the Levene test. In all cases, the level of significance α ≤ was 0.05, and the Confidence Interval (CI) was 95%.

Before analyzing the data obtained in the Ryff Psychological Well-being Scale, a reliability analysis was performed. The data obtained for the different scales showed a Cronbach internal consistency coefficient between 0.83 (self-acceptance) and 0.68 (personal growth). Therefore, the dimensions of the psychological well-being scale showed an acceptable internal consistency since the coefficients obtained were close to the results obtained in the validation of the instrument for the Spanish population [30].

## 3. Results

The descriptive results of the dimensions of the Ryff Psychological Well-being Scale show similar means between the two study groups (see Table 2). The dimension most valued by participants in both the experimental and comparison groups was positive relations. Note that this dimension refers to the capacity of participants to maintain social and emotional support for the provision of resources, especially for older people who live alone. Consequently, the results are consistent with those described in other studies that identify social support as one of the most valued indicators of the subjective well-being of older adults [8]. No significant differences were found between groups in the assessment of the separate dimensions of the Psychological Well-being Scale.

The descriptive results of daily memory evaluated by means of the RBMT test show that G1 obtained scores within the level of normality compared to G2, whose scores were identified within a weak memory level. The comparative data between the groups show that G1 obtained better results (see Table 3). The Stroop test, which measures the attentional level of the participants, again shows that the scores are within the normal range in both groups, but significant differences are shown in favor of the group that performed the computerized version of the memory training program (see Table 3). Finally, G1 has a significant advantage in the task of remembering words (see Table 3).

The descriptive results of the location of sources show that in G1, the most active areas in the baseline are located in the left hemisphere in the beta, alpha, and theta bands, and in the delta band, the active areas are located in the right hemisphere. In particular, the beta band is activated in the occipital and temporal areas, while the alpha band is activated in the inferior frontal area and precentral and temporal areas. In the theta band, the most active areas were located in the inferior and medial frontal areas, the insula, and the precentral area. In the delta band, the most active areas were the inferior frontal area, superior temporal area, and insula. In G2, bilateral areas were activated in the superior right and left temporal areas, and in the right hemisphere, the inferior frontal, precentral, and insula areas were activated. In the alpha band, the active areas were lingual and precursor. In addition, in the right hemisphere, the medial and superior occipital areas and the areas of the superior parietal lobe were activated. In the theta band, the active areas were the temporal, insula, lingual, occipital pole, and lower frontal area. In the right hemisphere, the active areas were the orbitofrontal medial cingulate, medial frontal, and cuneus. In the delta band, the active areas were the temporal, insula, precentral region, and lower frontal areas (see Table 4 and maps of Figure 1).

On the other hand, during the execution of the word memorization task, the most active areas in G1 were located in the right hemisphere, more specifically, in the superior frontal region and the medial temporal areas in the beta band. In the alpha band, the active areas were the inferior temporal and medial areas, and the occipital-temporal zone. In the theta band, the active areas were located in the inferior, superior, and medial temporal areas. In the delta band, the active areas were the medial and superior temporal areas, and the precentral area. The active areas during the memorization of words in G2 activation were bilateral. In the beta band, the cingulate area and medial temporal area were activated. In the alpha band, the active areas were the lingual area and curneus. In the theta band, the active areas were the superior temporal, medial, and supramarginal regions. In the delta band, the active areas were located in the occipitotemporal, temporal, and precentral regions.

The comparative results show greater activation in the alpha, theta, and beta bands in G2 than in G1. There are fewer significant differences in the slow bands (beta and alpha) than in the fast bands (theta and delta). The increased activation during verbal memory processing is located in the anterior and posterior areas. Consequently, it would appear that participants in G2 used neural polarization as a compensation strategy to optimize their performance [43] (See Table 4 and maps of Figure 2).

## 4. Discussion

This study evaluated psychological well-being, cognitive function, and cerebral processes in two groups of older people participating in the Memoria Mejor program. G1 completed exercises from the computerized program and G2 completed them by paper-and-pencil. The results in the psychological well-being scores show that there are no significant differences between people who carry out the program by paper-and-pencil versus on the computer. On the other hand, the objective results of well-being measured by the cognitive tests of attention, daily memory, and word list memory show significantly better scores in G1 versus G2. Similarly, the objective results at brain level show significant differences in the behavior of the beta, alpha, theta, and delta bands, depending on the support (computer vs. paper-and-pencil) of the Memoria Mejor program. That is, the computerized format of this training program contributes more effectively to achieving the objective criteria of successful aging.

### 4.1. Psychological Well-Being

Scores on the scale of psychological well-being were slightly higher in the group using the computerized program than in the group using pencil-and-paper. Although the results do not show significant differences between the groups, it is worth commenting on the relevance that “perceived social support” has for both groups [44,45]. In this sense, it should be noted that this same dimension is the leading indicator of the quality of life of older people with diseases that limit the functionality of their daily lives [46]. Since the participants of our study are healthy older people that have preserved their autonomy, this coincidence could be explained by the fact that social support provides care and protection for older people with health problems and allows healthy older adults to perceive themselves as having greater control and feeling more productive [47]. In addition, there is evidence that participants who perceive more social support resources tend to increase their social interactions and feel more socially integrated, thereby decreasing their risk of cognitive deterioration [48,49]. Regarding the assessment of the dimensions of psychological well-being of study participants living alone or with other people, perceived social support is again the most valued dimension. This difference can be explained by the fact that people who live alone perceive less social support when coping with psychosocial stress [46]. That is, living alone would not affect psychological well-being by itself, but the subject’s subjective evaluation of their own personal resources and environment would [50].

#### 4.1.1. Cognitive Processes

In relation to the objective criteria of successful aging evaluated by the cognitive processes of attention and daily memory, the results show that although both groups show scores within normal ranges, the scores are significantly better in G1 for both cognitive measures. This finding is consistent with systematic reviews carried out using computerized cognitive training showing that training improves cognitive performance in older people, regardless of their familiarity with technology [24,51]. Furthermore, there is sufficient evidence that computerized training programs improve the cognitive domains both in older people with normal aging and with deterioration [52,53]. However, given that this study does not have comparative data for impaired and unimpaired groups, the cognitive benefits of this type of program need to be investigated further. In addition, the literature does not have studies with scientific and methodological criteria that directly compare paper-and-pencil programs with computerized programs [14,51]. Most of these studies analyze the effect of computer programs by comparing the effect of virtual training between groups that receive training and control groups that do not receive any training [20]. In this study, given that the two groups use the same Memoria Mejor program but with different methods (classic versus computerized) and that G1 obtained better cognitive scores, we hypothesize that the computer environment contributes to a greater attentional effort and greater work memory load. In particular, the use of the keyboard and the mouse requires a performance in itself that entails supplementary instructions as opposed to classic programs where the environment is simply paper-and-pencil.

#### 4.1.2. Brain Band Activity

The objective criteria of successful aging associated with the modulation of brain band activity show significant differences between the groups in the basal state. In Group 1, activity is concentrated in the left hemisphere in the beta, alpha, and delta bands and the right hemisphere in the theta band, as expected in healthy older people (see Figure 1 and Figure 2). On the other hand, Group 2 shows bilateral activation in all bands (see Figure 1 and Figure 2), which shows a pattern closer to the cerebral decline of aging. The cognitive neuroscience of aging explains the idiosyncrasy of the brain aging of Group 2 by “noise” (or neuronal “chatter”) that progressively increases as the brain ages [22,54]. Increasing noise in the aging brain is associated with the degradation of interaction with the environment, slowdown in the speed of processing, and a decreased efficiency in the performance of working memory. Noise especially affects the “higher levels” of the brain system that are the first to be functionally disconnected when the age-related decline appears and subsequently weakens the feedback between the anterior-posterior system [20,55]. Regarding the modulation of the activity of the frequency bands during the task of memorizing words, both groups adopt a compensation pattern in response to the demand of the verbal task. Particularly, in G1, the compensation consists of an over-activation of the frequency bands in the right hemisphere. This type of compensation is explained by the literature on the neuroscience of aging with the additional effort required by cognitive tasks of high demand. Note that the proposed memorization task is associated with temporal areas of the left hemisphere, and the participants show contralateral activity in only the right hemisphere. This type of contralateral activation is also found in young adults when cognitive tasks are highly demanding. On the other hand, in Group 2, the pattern of cerebral compensation is bilateral, which is typically associated with aging brains. The reduction of hemispheric asymmetry in older people typically manifests itself in tasks of operational memory with verbal and spatial material. Although the processing of linguistic stimuli exhibits left lateralization and the processing of spatial stimuli exhibits right lateralization, aging brains show a pattern of bilateral activation in both tasks. In addition, the absence of hemispheric asymmetry has also been identified in tasks that require both superficial (perceptive) and deep (semantic) processing. In short, older people with aging brains tend to activate both cerebral hemispheres, regardless of the type of material, the complexity of the task, or the complexity of the stimulus.

Although both groups present a compensation pattern based on the demand of the task, in Group 1, the brain pattern is associated with the demand of the task, while in Group 2, the pattern of compensation is typically similar to that of participants with neuro-cognitive deficits associated with age. Paper-and-pencil and computerized training programs aim to renormalize the neurological processing of aging brains by performing cognitive attention, memory, and processing speed exercises. These processes are involved in the activation of coordination between higher levels and primary or automated regions. The premise on which computer programs are based is that a brain with noise is substantially reparable through the resources available to the brain system itself to restore its weakened or lost capacities [22]. The brain is plastic throughout a person’s life, and training acts not at the palliative level, but the curative level [56]. The restoration of the normalization of brain modulation is measured in terms of less activation and greater cognitive efficacy. Following this argument, G1 shows better results since fewer active areas were located and there was a higher cognitive performance (see Figure 1 and Figure 2). Group 2 shows greater cerebral activation in contralateral areas in the slow bands (beta and alpha) and bilateral activation in fast bands (theta and delta) with worse cognitive results. Keep in mind that while Group 1 remembered 6.56 words, Group 2 remembered 5.12. Therefore, the computerized version of the program fulfills the premise that it is possible to restore cognitive and cerebral functioning, and consequently, it is better suited to achieving the objective criteria of satisfactory aging [57].

## 5. Conclusions

Regarding the assessment of the dimensions of psychological well-being of study participants living alone or with other people, perceived social support is again the most valued dimension. This difference can be explained by the fact that people who live alone perceive less social support when coping with psychosocial stress [46]. That is, living alone would not affect psychological well-being by itself, but the subject’s subjective evaluation of their own personal resources and environment would [50].

## Figures and Tables

**Figure 1 ijerph-16-00191-f001:**
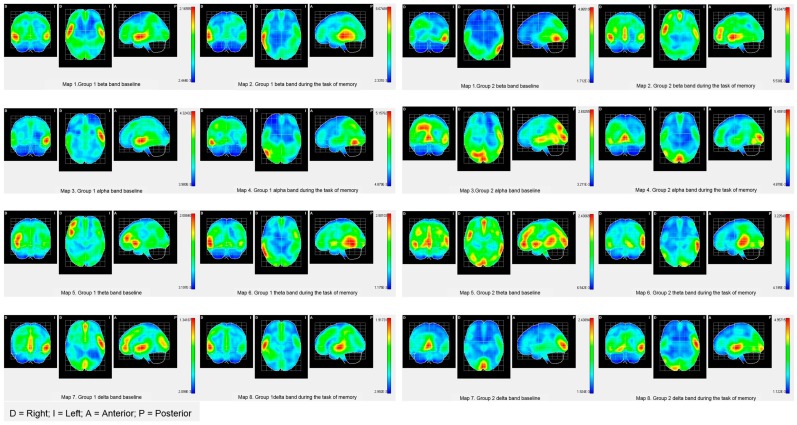
Descriptive maps of brain activity of G1 and G2.

**Figure 2 ijerph-16-00191-f002:**
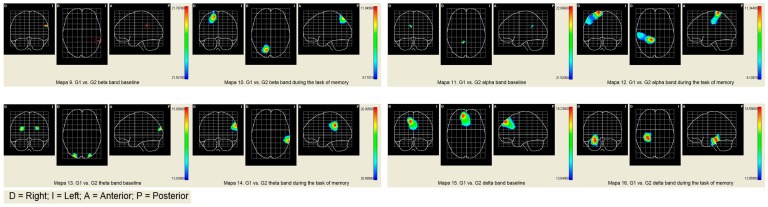
Comparative maps of brain activity of G1 and G2.

**Table 1 ijerph-16-00191-t001:** Demographic variables of the training groups.

Subjects	Group 1	Group 2
***N* = 54**	26	28
**Age**	73.61 (±3.32)	74.25 (±1.9)
**Sex**		
Male	10	11
Female	16	17
**Living arrangement**		
Alone	10	11
Accompanied	16	17
**Education level**		
University	3	0
High school	4	6
Primary school	19	22
**Job**		
Civil servant	6	7
Businessman	3	5
Farmer	4	4
Housewife	13	12

**Table 2 ijerph-16-00191-t002:** Mean values (X) and standard deviation (SD) of dimensions of the Psychological Well-being Scale.

Dimensions	Group 1		SD	Group 2		SD
Self-Acceptance	27.53	±	4.37	26.13	±	2.57
Positive Relations	34.71	±	3.00	32.00	±	5.33
Alone	32.98	±	3.58	30.22	±	3.80
Accompanied	35.78	±	2.52	33.14	±	2.35
Autonomy	25.29	±	4.22	24.47	±	5.80
Environmental Mastery	28.94	±	3.38	26.00	±	4.47
Personal Growth	31.06	±	3.40	30.60	±	4.32
Purpose in Life	28.00	±	3.55	26.67	±	3.95

**Table 3 ijerph-16-00191-t003:** Mean values (X) and standard deviation (SD) of cognitive measures of the two study groups.

Cognitve Test	Group 1			Group 2			Group 1 vs. Group 2
	Mean		SD	Mean		SD	*P*
Atention	3.22	±	4.35	−2.90	±	5.47	*p* = 0.001
Everyday memory	11.23	±	0.24	−9.45	±	1.13	*p* = 0.002
Word List Recall	6.56	±	1.42	5.12	±	1.38	*p* = 0.014

**Table 4 ijerph-16-00191-t004:** Locations give Brodmann area in the two study groups.

Brain Activities	PALA	BA	X	Y	Z	T2 Hotelling	Mean	
							G1	G2
**Beta band**								
Baseline	Left Angular gyrus	39	140	104	144	9.1357 **	0.3452	0.3662
Task of memory	Right superior Parietal Lobe	7	76	126	164	11.0486 *	0.0688	0.2250
**Alpha band**								
Baseline	Right Cingulate Region	7	76	100	136	22.0856 ***	0.1019	0.1119
Task of memory	Postcentral area right	3	76	148	128	11.3437 *	0.2848	07802
**Theta band**								
Baseline	Superior left occipital area	19	120	96	184	15.6583 **	0.1442	0.4200
Task of memory	Left Occipital Pole	18	108	52	192	9.6772 *	0.4046	0.4883
	Occipitotemporal area Lateral left	18	108	56	168	9.1443 *	0.7235	0.7850
	Lingual Area Left	18	108	60	180	9.39411 *	20.845	25.436
**Delta band**								
Baseline	Right Lateral Orbitofrontal Region	11	72	48	76	11.37077 *	0.0926	0.1901
Task of memory	Right parahipocampal area	30	64	52	116	18.3420 **	0.4071	0.4257
	Right lateral Occipitotemporal area	30	64	52	120	18.5904 **	0.3481	0.3735
	Hippocampal area right	30	64	52	112	17.9080 **	0.4427	0.4594

PALA = Probabilistic Atlas of location areas. BA = Brodmann areas. x, y, z = identification of coordinates. Statistical significance * *p* < 0.05; ** *p* < 0.01 and *** *p* < 0.001.

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
