# Peer review of "Evaluating Successful Aging in Older People Who Participated in Computerized or Paper-and-Pencil Memory Training: The Memoria Mejor Program"

_ijerph, 2019, doi:10.3390/ijerph16020191_

Round 1

Reviewer 1 Report

I feel the first sentence of the Introduction – as written - is not correctly written and is misleading.  Successful aging does NOT include objective criteria.  Successful aging is pursued by an older individual and evaluative criteria are not part of their world or their concerns.  Rather, the first sentence should say “The evaluation of successful aging includes objective criteria to measure levels of functional capacity………..”  The first sentence of the abstract is similarly wrongly written. 

Line 17 – change adults to adult

Line 36-38.  The assertion that “A large group of researchers agree…” must include a citation of sources where this assertion has been made by several researchers.

Line 39-41.  “Wellness is defined as…..” needs a citation listing where in the literature wellness has been defined as such.

Starting on line 44.  I take strong exception with these authors’ assertions that “The subjective keys to aging in a satisfactory manner are directly related to personal evaluations of the value or utility of the activities…..”   Extensive literature has looked at successful aging from the standpoint of health and wellness, psychological well-being, community connections, appropriate housing, familial relationships, economic security, and a variety of other conditions.  The pursuit of fulfilling activities is only of many concepts related to successful aging.

A major flaw in the reporting of data is that on lines #110 – 111, the authors state: “Group 1 (G1) carried out the computerized program and Group 2 (G2) did the program using paper and pencil.”  Then, in Discussion lines #7-8, the authors state, “G1 completed exercises from the program by paper and pencil and G2 completed them on computer.”  This significant discrepancy needs to be flushed out to ensure data have been presented correctly. 

Methods

Why and how the group was reduced from 82 to 54 participants needs to be explained.

The age of participants should be included, as it relates to their ability to achieve successful aging.  The older one becomes, the more difficult it becomes to achieve what may be defined as “successful aging” on a variety of indicators. 

So, had participants been participating in a memory training program for three years before this study began?  What the three years of this program entailed, and what exactly is the “Memoria Mejor” needs to be more thoroughly explained.  

Discussion

I find it very difficult to believe the authors’ assertion that “this is the first study that evaluates satisfactory aging through [both] subjective and objective well-being criteria.”  In the countless studies that look at aging and well-being across a wide variety of measures, it is hard for me to fathom that this is the first study that has used both subjective and objectives measures. 

In addition, this study did not measure satisfactory aging.  Rather it measured whether psychological well-being (as measured by the Ryff Scale), memory (as measured by the RBMT and WMS), and cognitive functioning (as measured by EEG) were impacted by whether participants had previously participated in a memory training program administered either by computer OR paper-and-pencil.  That is what this study measured and nothing more.  Satisfactory aging encompasses so much more than these particular constructs.  For this reason, I suggest that the article title be changed accordingly as well, perhaps “The evaluation of specific measure of successful aging in older people who had participated in two means of administering a memory program.”

The authors need to discuss shortcomings and limitations of the study within their Discussion.

Author Response

Dear reviewer,

First of all, I would like to thank you for your valuable comments that have helped to improve the writing of the manuscript. I have responded to each of your comments.

Reviewer 1

I feel the first sentence of the Introduction – as written - is not correctly written and is misleading.  Successful aging does NOT include objective criteria.  Successful aging is pursued by an older individual and evaluative criteria are not part of their world or their concerns.  Rather, the first sentence should say “The evaluation of successful aging includes objective criteria to measure levels of functional capacity………..”  The first sentence of the abstract is similarly wrongly written. 

Answer 1: Following the instructions of the reviewer, the sentence has been modified (see lines 10 to 11 of the abstract and lines 31 to 32 of the section of the Introduction)

Line 17 – change adults to adult

Answer 2: Following the instructions of the reviewer, the word has been changed ((see line 20)

Line 36-38.  The assertion that “A large group of researchers agree…” must include a citation of sources where this assertion has been made by several researchers.

Answer 3: The citation has been implemented (see page 38)

Line 39-41.  “Wellness is defined as…..” needs a citation listing where in the literature wellness has been defined as such.

Answer 4: The citation has been implemented (see page 41)

Starting on line 44.  I take strong exception with these authors’ assertions that “The subjective keys to aging in a satisfactory manner are directly related to personal evaluations of the value or utility of the activities…..”   Extensive literature has looked at successful aging from the standpoint of health and wellness, psychological well-being, community connections, appropriate housing, familial relationships, economic security, and a variety of other conditions.  The pursuit of fulfilling activities is only of many concepts related to successful aging.

Answer 5: In response to the reviewer's comment, the sentence has been rewritten. Now the sentence says: "The pursuit of fulfilling activities is included among the concepts related to successful aging" (see line 45). That is, the phrase has been fixed so that "the usefulness of the activities" is a variable among o41thers that contributes to the satisfaction of aging.

A major flaw in the reporting of data is that on lines #110 – 111, the authors state: “Group 1 (G1) carried out the computerized program and Group 2 (G2) did the program using paper and pencil.”  Then, in Discussion lines #7-8, the authors state, “G1 completed exercises from the program by paper and pencil and G2 completed them on computer.”  This significant discrepancy needs to be flushed out to ensure data have been presented correctly. 

Answer 6: The error has been corrected (see line 115)

 Methods

Why and how the group was reduced from 82 to 54 participants needs to be explained.

Answer 7: The Methods section has been rewritten to improve its comprehensibility. This question has been answered (see lines 111-113).

The age of participants should be included, as it relates to their ability to achieve successful aging.  The older one becomes, the more difficult it becomes to achieve what may be defined as “successful aging” on a variety of indicators. 

Answer 8: the age has been included in the table 1.

So, had participants been participating in a memory training program for three years before this study began?  What the three years of this program entailed, and what exactly is the “Memoria Mejor” needs to be more thoroughly explained.  

Answer 9: The method section has been rewritten to dispel doubts and confusion.

Discussion

I find it very difficult to believe the authors’ assertion that “this is the first study that evaluates satisfactory aging through [both] subjective and objective well-being criteria.”  In the countless studies that look at aging and well-being across a wide variety of measures, it is hard for me to fathom that this is the first study that has used both subjective and objectives measures. 

Answer 10: The vehemence of the first sentence of the discussion has been eliminated.

In addition, this study did not measure satisfactory aging.  Rather it measured whether psychological well-being (as measured by the Ryff Scale), memory (as measured by the RBMT and WMS), and cognitive functioning (as measured by EEG) were impacted by whether participants had previously participated in a memory training program administered either by computer OR paper-and-pencil.  That is what this study measured and nothing more.  Satisfactory aging encompasses so much more than these particular constructs.  For this reason, I suggest that the article title be changed accordingly as well, perhaps “The evaluation of specific measure of successful aging in older people who had participated in two means of administering a memory program.”

Answer 11: The suggestion to change the title proposal has been followed.

The authors need to discuss shortcomings and limitations of the study within their Discussion.

Answer 12: The limitations of the study have been included in the final part of the discussion.

Reviewer 2 Report

why statistical analysis section is in the results? i think that the groups differs among them. please report age and p value at baseline test of normality? must be done

Author Response

Dear reviewer,

First of all, I would like to thank you for your valuable comments that have helped to improve the writing of the manuscript. I have responded to each of your comments.

why statistical analysis section is in the results? I think that the groups differ among them. please report age and p value at baseline test of normality? must be done 

answer 1: The description of the statistical analyzes has been removed from the results section.  

The age variable is included in table 1.

The Kolmogorov Smirnov normality test was carried out with the Lilliefors correction for all the variables obtaining a value p> 0.05

Reviewer 3 Report

Thank you for allowing me to read your interesting paper.

Evaluation of a memory program, using both subjective and objective measures, merits attention. I enjoyed reading your paper. However, before considering for publication, I have some concerns and questions to be addressed by authors.

Major concerns,

Please clarify the reason (or rationale) for inclusion criteria of the study participants. Why participants should be between 3 and 5 years of retirement? Why 3 year training in Memoria Major program necessary? Why participants who answered “learn new skills and abilities” were selected? Such kind of information is necessary in Methods section.

How was the ethical consideration for this study? Did you obtain any approval from the ethical committee? If so, please specify.

Authors mentioned about the study results. Authors based their conclusion on mere comparison between group 1 and 2. In addition, demographic variables in the table 1 shows that group 1 has university educated participants. Moreover, they were slightly younger than group 2 participants. As sample size is small, slight difference might influence the result. Given this, I suggest that authors compare before and after the program result, if possible. Otherwise, we are not sure if the difference observed in this study is a program effect or not. There might be other factors which are likely to affect the result. In sum, based on your study results, conclusion part in the abstract are not convincing enough.

Table 4 is unclear. What this table represents? We are not sure which figure is for group1 and 2. Please show results for group 1 and 2 separately.

The description about “Memoria Major” program is insufficient (see page 4). Since this is an intervention study, more detailed description about the program might help. For example, we are not sure if 75 minute session is conducted weekly or monthly? No information about the setting. Community center or university? Who conducted the session?. etc...

Discussion section as a whole is not well organized. Please re-organize the section based on the study results and mention studies that are only relevant to your results. Providing sub-headings might help organizing the section.

Minor concerns

Authors mentioned that Ryff Psuchological Well-being Scale has six dimensions (p4). However, in table 2, eight dimensions are shown. Please clarify.

I do not understand the sentence “the dimension most valued by participants… was autonomy (p6, line 221-2). In table 2, autonomy has a value of 25.29 and 24.47 for group1 and 2 respectively, but such figures do not tell us that autonomy was most valued.

Table 3 is ugly.

Please check the reference again. For example, the information on reference 14 is not accurate.

Author Response

Dear reviewer,

First of all, I would like to thank you for your valuable comments that have helped to improve the writing of the manuscript. I have responded to each of your comments.

Please clarify the reason (or rationale) for inclusion criteria of the study participants. Why participants should be between 3 and 5 years of retirement? Why 3 year training in Memoria Major program necessary? Why participants who answered “learn new skills and abilities” were selected? Such kind of information is necessary in Methods section.

Answer 1: The method section has been rewritten to dispel doubts and confusion (See lines 101 to 104).

How was the ethical consideration for this study? Did you obtain any approval from the ethical committee? If so, please specify.

Answer 2: A new paragraph has been written explaining the procedures of the ethical committee that were followed in this study (see lines 116 to 120)

Authors mentioned about the study results. Authors based their conclusion on mere comparison between group 1 and 2. In addition, demographic variables in the table 1 shows that group 1 has university educated participants. Moreover, they were slightly younger than group 2 participants. As sample size is small, slight difference might influence the result. Given this, I suggest that authors compare before and after the program result, if possible. Otherwise, we are not sure if the difference observed in this study is a program effect or not. There might be other factors which are likely to affect the result. In sum, based on your study results, conclusion part in the abstract are not convincing enough.

Answer 3: The limitations of the study have been included in the final part of the discussion.

Table 4 is unclear. What this table represents? We are not sure which figure is for group1 and 2. Please show results for group 1 and 2 separately.

Answer 4: The data corresponding to each group has been clarified in table 4.

The description about “Memoria Major” program is insufficient (see page 4). Since this is an intervention study, more detailed description about the program might help. For example, we are not sure if 75 minute session is conducted weekly or monthly? No information about the setting. Community center or university? Who conducted the session?. etc...

Answer 5: New information has been implemented in the Memoria Mejor program section in relation to the organization of training sessions and trainers’ training (see lines 155 a 170).

Discussion section as a whole is not well organized. Please re-organize the section based on the study results and mention studies that are only relevant to your results. Providing sub-headings might help organizing the section.

Answer 6: The discussion has been ordered and sub-headings are provided.

 Minor concerns

Authors mentioned that Ryff Psychological Well-being Scale has six dimensions (p4). However, in table 2, eight dimensions are shown. Please clarify.

Answer 7: Indeed, the Ryff scale consists of six dimensions. Table 2 presents the results of the six dimensions of G1 and G2. Additionally, in the "positive relations" dimension, the average of the participants who live alone and accompanied are included.

I do not understand the sentence “the dimension most valued by participants… was autonomy (p6, line 221-2). In table 2, autonomy has a value of 25.29 and 24.47 for group1 and 2 respectively, but such figures do not tell us that autonomy was most valued.

Answer 8: The error has been corrected. The most valued dimension was "positive relations" (see lines 24 to 27 of the discussion).

Table 3 is ugly.

Answer 9: Table 3 has been corrected.

Please check the reference again. For example, the information on reference 14 is not accurate.

Answer 10: The references have been revised again, and reference 14 has been corrected.

Reviewer 4 Report

Please see comments in the attached.

This study used two different approaches (pencil-paper and computer-based) memory program to examine the effect of brain activity, cognitive function, and psychological well-being on older adults. The computer interface is getting popular in nowadays, and the effectiveness should be examined. The strength of this study is that the EEG technique in addition to questionnaire were applied to examine the brain functioning in the test, and it showed the different active regions in the two groups. The findings provide information about the differences in the brain. Here are some comments I'd like to provide for authors' reference.
1. It seems the psychological well-being and cognitive function were compared between two groups after the intervention, but there were no pre-test data of the two groups. From the characteristics of the participants,
Group 1 seemed to be highe educated, and more likely to have company. The differences between 2 groups may come from the basic differences of the participants, but not from the intevention effect. If the pre-test data were available, it should be compared before and after of two groups, but not directly comparing post-test of 2 groups. Paired t test or further GEE analysis by controlling the covariates such as education and social support should be considered.  In case there were no pre-test data, this would be a major drawback of the study design, and
at least it should  be addressed in the limitation.

2.Page 12, Table 4: In the text it indicates Figure 4, which should be corrected. And the content of the note (such as significance, abbreviations) should be put in the note under the table, but not in the table title.

3. The authors compared the EEG brain activity of two groups, and concluded that Group 2's brain is more aging. If the more aging brain compensate the working memory and other brain tasks, to make the final memory performance very close, then that means both intervention approaches work for memory improving. The implication of training and practicing brain activity for aging people should be emphasized and encouraged.

Author Response

Dear reviewer,

First of all, I would like to thank you for your valuable comments that have helped to improve the writing of the manuscript. I have responded to each of your comments.

1. It seems the psychological well-being and cognitive function were compared between two groups after the intervention, but there were no pre-test data of the two groups. From the characteristics of the participants,
Group 1 seemed to be high educated, and more likely to have company. The differences between 2 groups may come from the basic differences of the participants, but not from the intervention effect. If the pre-test data were available, it should be compared before and after of two groups, but not directly comparing post-test of 2 groups. Paired t test or further GEE analysis by controlling the covariates such as education and social support should be considered.  In case there were no pre-test data, this would be a major drawback of the study design, and at least it should  be addressed in the limitation. 

Answer 1: It is a retrospective study, and pre-test evaluation data is not available. The limitations of the study have been included in the final part of the discussion. The method section has been rewritten to dispel doubts and confusion (See lines 109 to 111).

2.Page 12, Table 4: In the text it indicates Figure 4, which should be corrected. And the content of the note (such as significance, abbreviations) should be put in the note under the table, but not in the table title. 

Answer 2: Table 4 has been corrected.

3. The authors compared the EEG brain activity of two groups, and concluded that Group 2's brain is more aging. If the more aging brain compensate the working memory and other brain tasks, to make the final memory performance very close, then that means both intervention approaches work for memory improving. The implication of training and practicing brain activity for aging people should be emphasized and encouraged.

Answer 3:  Although both groups present a compensation pattern to the demand of the task, in Group 1 the brain pattern is associated with the demand of the task, while in Group 2 the pattern of compensation is typically similar to that of participants with neuro-cognitive deficits associated with age. Therefore, brain activity in the G1 could be said to be normalized when compared with younger age groups, compared to that of G2 in which both the baseline and the process of word memorization activates compensation patterns. Therefore, the computerized program seems more recommendable since it does not act at the palliative level but at a curative level as compared to the paper and pencil program which seems to act in palliative terms (See lines 97 to 114)

Round 2

Reviewer 2 Report

the manuscript has been improved

Author Response

The English language and style are fine/minor spell had been checked 

Reviewer 3 Report

Thank you for re-submitting the paper. It seems that most concerns are addressed. However, I found further concerns to be addressed.

1. Please use the term correctly.  For example, the use of "inclusion criteria" in method section is confusing. In this case, authors just described the "sample characteristics", not "criteria". 

I do not understand the sentence in line 102-103, p3.

2. Check the figure

Table 1  Why there are 54 people in each group?  in "cohabitation", (by the way, this should be "living arrangement" .."cohabitation" means living with someone else) .

Table 2 Are figures in positive relations, the means of those living alone and living with someone else combined? Then why such a discrepancy in fugures such as 34.71 for total and 24.36 for alone, 26.43 for accompanied? I suggest to chek all the figure in the table again.

Overall, some sentences are redundant and unclear. I strongly recommend editing service by native English speaker.

Author Response

We appreciate the comments of the reviewer since they have been of great help in improving the clarity and precision of the manuscript.

Happy Holidays

Please use the term correctly.  For example, the use of "inclusion criteria" in the method section is confusing. In this case, authors just described the "sample characteristics", not criteria". 

Following the instructions of the reviewer, inclusion criteria have been included as characteristics of the sample and have been implemented as an inclusion criterion. Participants in the Memoria Mejor training program used both a paper-and-pencil and computerized (see lines 166-177)

I do not understand the sentence in line 102-103, p3.

The sentence has been corrected (see lines 103-105)

2. Check the figure

Table 1  Why there are 54 people in each group?  in "cohabitation", (by the way, this should be "living arrangement" .."cohabitation" means living with someone else).

It now says "living arrangement" instead of "cohabitation"

Table 2 Are figures in positive relations, the means of those living alone and living with someone else combined? Then why such a discrepancy in figures such as 34.71 for total and 24.36 for alone, 26.43 for accompanied? I suggest to check all the figure in the table again.

The data referring to "social support" in table 2 have been corrected

Overall, some sentences are redundant and unclear. I strongly recommend editing service by native English speaker.

The manuscript has been reviewed again by a native expert. Changes have been made especially in the discussion and conclusion section.

Reviewer 4 Report

The authors have revised and replied to the comments. Although the limitation of no pre-test before the intervention is a major issue in an experimental study to confirm the effect, there are some creative findings from this study, and the limitation has been added in the last paragraph. 

Author Response

The manuscript has been reviewed again by a native expert. Changes have been made especially in the discussion and conclusion section